# Utilization of Gasification Coarse Slag Powder as Cement Partial Replacement: Hydration Kinetics Characteristics, Microstructure and Hardening Properties

**DOI:** 10.3390/ma16051922

**Published:** 2023-02-25

**Authors:** Kuizhen Fang, Dongmin Wang, Yue Gu

**Affiliations:** School of Chemical and Environmental Engineering, China University of Mining & Technology-Beijing, Beijing 100083, China

**Keywords:** gasification coarse slag powder, supplementary cementitious material, dissolution characteristics, hydration kinetics, degree of reaction

## Abstract

Coal gasification coarse slag (GFS) is a byproduct of coal gasification technology, which contains abundant amorphous aluminosilicate minerals. GFS has low carbon content, and its ground powder has potential pozzolanic activity, which can be used as a supplementary cementitious material (SCM) for cement. Herein, GFS-blended cement was studied in terms of ion dissolution characteristics, initial hydration kinetics, hydration reaction process, microstructure evolution process, and the development of the mechanical strength of their paste and mortar. Enhanced alkalinity and elevated temperature could increase the pozzolanic activity of GFS powder. The specific surface area of GFS powder and its content did not change the reaction mechanism of cement. The hydration process was divided into three stages: crystal nucleation and growth (NG), phase boundary reaction (I), and diffusion reaction (D). A higher specific surface area of the GFS powder could improve the chemical kinetic process of the cement system. The degree of reaction of GFS powder and blended cement had a positive correlation. A low GFS powder content (10%) with a high specific surface area (463 m^2^/kg) showed the best activation in cement as well as improving the late mechanical properties of cement. The results show GFS powder with low carbon content has the application value as SCM.

## 1. Introduction

Coal has played an important role in the development of human civilization, but it has also brought about serious damage to the ecological environment of Earth, in the forms of acid rain, a hole in the ozone layer, and the greenhouse effect [1,2,3]. Therefore, in the realization of a sustainable development strategy, coal must be used cleanly. In recent years, with the clean utilization of coal and the rise of a new chemical industry based on coal, especially in the increase in coal-to-oil and coal-to-gas projects, there have been rapid developments in the coal gasification industry in China [4,5,6,7]. As these projects generate a large amount of coal gasification slag, the stockpiles of this material are growing in size every year. According to statistics, around 30 million tonnes of coal gasification slag are produced annually in China alone [8,9,10]. Coal gasification slag is a black powdery material that is rich in aluminosilicate minerals and has 5–40% carbon residue. Its storage occupies a large amount of available land and seriously affects the surrounding ecological environment [11,12]. Therefore, coal gasification slag must play a positive role in the clean utilization technology of coal in China. To this aim, the disposal and utilization of this waste have become of major concern [13,14,15].

Figure 1 shows an overview of the processing of coal gasification slag [13]. Coal gasification slag is divided into gasification fine slag (GFFS) and gasification coarse slag (GFS), of which the GFS content is around 60–80% [16]. Due to the processes and reaction conditions, the carbon content of GFFS is high. Research on this material has been focused on carbon-ash separation, boiler combustion, carbon-based adsorbents, and the preparation of high-value-added products [17,18,19,20,21,22,23]. As the carbon content of GFS is low, there is the opportunity for it to be used in the field of building materials, such as in burning cement clinker, preparing adsorption materials, and preparing fine sand to replace the crushed stone layer or the fine aggregate of the pavement base material [24,25,26,27,28]. GFS, as opposed to GFFS, is suitable for use as a supplemental material for cement due to its chemical and mineral-phase composition.

In the field of building materials, the production of cement requires the use of a large amount of limestone and clay and produces a high volume of greenhouse gases, mainly in the form of carbon dioxide [29]. According to the Statistical Review of World Energy released by British Petroleum (BP) in 2019, global carbon emissions were estimated to be around 3.416 billion tonnes. Of these, the carbon emissions of the cement industry account for around 7% of the global anthropogenic greenhouse gas (GHG) emissions [30,31,32]. To reduce carbon dioxide emissions and the consumption of natural resources during cement production, supplementary cementitious materials (SCMs) are used to partially replace cement, such as fly ash, blast-furnace slag, and metakaolin [33,34,35]. However, in recent years, the demand for cement has been huge, and the annual output of SCMs is relatively limited. In addition, SCMs have problems such as poor stability and uneven distribution of resources, which make them difficult to be widely used to replace cement in large quantities. Therefore, with the requirement that they do not significantly affect the performance of cement and concrete, SCMs that are rich in resources should be sought that can be used to partially replace cement, which provides an opportunity for the utilization of GFS resources in building materials.

Compared with other mineral admixtures, the composition of GFS is closer to that of fly ash (as shown in Figure 2), with more SiO_2_ and Al_2_O_3_ and amorphous phase materials. The potential pozzolanic activity and particle filling effect of GFS powder can play an important role in cement-based materials. In recent years, research has been focused on the properties of coal gasification slag and the impact it has on the performance of cement and concrete [36,37]. Feng et al. [24,38] applied the decarburized gasification slag to cement and mortar and found that the GFS powder could delay the hydration heat release and setting of cement. Mechanical activation could promote the improvement of the activity of GFS. Bo et al. [36] also found that the improvement of fineness could promote the better nucleation and pozzolanic effect of GFS powder in cement. Moreover, active amorphous SiO_2_ and Al_2_O_3_ play a positive role in promoting the reaction between GFS powder and cement, especially in the performance of the later mechanical properties of cement paste and mortar [13,39]. Currently, limited research studies have been conducted, and the reported mechanistic and experimental studies that have been conducted on this topic are not comprehensive. To date, there has been an insufficient number of theoretical studies published on the utilization of GFS in building materials. Therefore, it is necessary to systematically study the reactivity characteristics of GFS and the influence that it has on the mechanism of the hydration of cement and the hardening of the material. In particular, to deepen the research on the influence mechanism of GFS powder on cement hydration kinetics and quantify the degree of reaction of GFS powder in the cement hydration process. Ultimately, through the analysis of experimental data and theoretical research, the reaction characteristics of GFS in cement can be further understood. This will help to carry out further activation modification research on GFS from the level of the reaction mechanism.

GFS powder was taken as the research object. The effects of GFS powder with different specific surface areas and its content in blended cement on the degree of hydration and hardening characteristics of cement were investigated, in terms of hydration kinetics characteristics, the hydration reaction process, the morphology and microstructure evolution of the hydration products, and the development in the mechanical strength of paste and mortar. The experimental results provide a theoretical basis for further study of the activation modification of GFS powder and its application in cement.

## 2. Materials and Methods

### 2.1. Materials

GFS produced in the Ningdong region of Ningxia, China, at a gasification temperature of 1400–1600 °C under a pressure of 4.2–4.5 MPa with a carbon content of <5% was used in the experiments. Reference cement (OPC) with a specific surface area of 368 m^2^/kg was used, which meets the requirements of GB 8076. The chemical compositions of the OPC and GFS are shown in Table 1. The GFS was ground in a ball mill to a fineness of 3, with specific surface areas of 391 m^2^/kg, 425 m^2^/kg, and 463 m^2^/kg, which are referred to as GFS-I, GFS-II, and GFS-III, respectively. Figure 3 and Figure 4 show the morphology and particle size distribution of GFS after grinding, respectively. The chemical composition of GFS mainly comprises SiO_2_ and Al_2_O_3_, as well as a small amount of Fe_2_O_3_ and CaO, making it a silica–aluminum raw material. The micromorphology of GFS is mainly composed of spherical SiO_2_ particles and irregular silica–aluminum particles, which after the grinding of the material adopt irregular block shapes.

Figure 5 shows the mineral-phase composition of the GFS powder, which mainly comprises a crystalline SiO_2_ phase and amorphous aluminosilicate minerals. Aluminosilicate minerals that exhibit a high degree of amorphousness (glass phase content is 88%), usually amorphous SiO_2_ and Al_2_O_3_, exhibit pozzolanic activity and play certain roles in the hydration and hardening process of cement. Figure 6 shows the Fourier-transform infrared (FTIR) spectrum of the GFS powder. Among them, the wide absorption band at 3442 cm^−1^ can be attributed to the vibration of hydroxyl groups and the peaks below 1300 cm^−1^ can be attributed to the vibrations of silicoxy and aluminoxy groups. The main peak in the spectrum at 1042 cm^−1^ may be caused by the asymmetric vibration of Si–O–Si or Si–O–Al bonds and the peak for Si–O at 797 cm^−1^ may be attributed to quartz, with the peaks at 465 cm^−1^ presumed to be due to the bending vibration of Si–O bonds. These results are consistent with the characteristics of the sample featuring a high Si and Al content, indicating that it exhibits a typical amorphous aluminosilicate structure [17,40]. The vitreous phase in GFS powder belongs to siliceous material, and the framework structure is mainly composed of SiO_4_ tetrahedrons joined by bridging oxygen (Si-O-Si). Compared with calcareous fly ash and granulated blast furnace slag powder, the framework structure of GFS is more difficult to depolymerize [41].

### 2.2. Methods

#### 2.2.1. Dissolution Testing of the GFS Powder

The concentration of ions in the sample was measured by inductively coupled plasma emission spectrometry (ICP-MS) [42], the sample preparation of which involved the following. Powdered GFS (3 g) was dissolved in deionized water (100 mL) or a certain concentration of a solution of sodium hydroxide (0.2 mol/L, 0.4 mol/L, 0.6 mol/L, NaOH) and stirred in a magnetic agitator under different temperatures (20 °C, 40 °C, 60 °C) for a specified time. After this process, the sample was centrifuged, the supernatant was collected, filtered through a 0.45μm filter and acidified with nitric acid (HNO_3_) to make the solution acidic for ICP-MS testing. The sample used in this experiment was GFS-III, as this sample exhibits the highest specific surface area.

#### 2.2.2. Preparation of Blended Cement and Mortars Containing the GFS Powder

Blended cement pastes were prepared using different GFS powder content (10%, 30%, 50%) in place of the cement. The water-to-ash ratio is 0.35. Samples were molded in prismatic casts with dimensions of 30 mm × 30 mm × 30 mm. Table 2 shows the specific mixing ratio of the pastes, where the samples were cured at 20 ± 1 °C and a humidity of 95% for different periods (3 d, 7 d, 14 d, 28 d, and 90 d).

The mortar samples were prepared according to the “Testing method for the strength of cement mortar” (GB/T 17671, Chinese standard method), with Table 2 showing the mixing ratios of the cement. After 24 h, the samples were demolded and then cured at 20 ± 1 °C and a humidity of 95% for different periods (7 d, 28 d, and 56 d) before testing their compressive and flexural strength properties.

#### 2.2.3. Performance Testing of the Blended Cement Containing GFS Powder

(1) Determination of the hydration heat

An eight-channel TAM air isothermal microcalorimeter was used to test the heat release properties of the samples, according to ASTMC 1679. The total masses of the samples were calculated based on the specific heat capacity of the reference sample (Quartz sand). The heat release rate and total heat release of blended cement containing GFS were tested at 25 °C over a period of 72 h, and the sample mixing ratio was shown in Table 2. The hydration kinetics parameters were calculated and fitted.

(2) Testing of the degree of hydration of the blended cement

The degree of hydration of the blended cement was evaluated in terms of its non-evaporated water content (chemically-bound water). Before taking measurements, the samples to be measured were crushed and immersed in anhydrous ethanol to prevent their hydration. Then, the samples were placed in an oven set at 65 °C to dry them to a constant weight. After grinding and sieving, 1–2 g of the samples were taken and burnt for 2 h at 1000 °C until a constant weight was reached. The remaining mass was weighed and the chemically-bound water content was calculated using the following Equation (1):(1)$$W_{n}=\frac{W_{1}{-W}_{2}}{W_{2}}-\frac{L_{\mathrm{SS}}}{{1-L}_{\mathrm{SS}}}$$
where *W_n_* is the non-evaporated water content of the sample in %, *W*_1_ is the mass of the sample in g before burning, *W*_2_ is the mass of the sample in g after being burnt at 1000 °C, and *L_SS_* is the percentage loss (%) of the cementitious materials on ignition.

(3) Testing of the degree of reaction of the GFS

The “Quantitative Determination of Cement Components” according to GB/T 12960 (a nitric acid selective dissolution Chinese standard method) was used as a method to test the degree of reaction of the GFS. The samples were prevented from being hydrated using anhydrous ethanol and were then dried in an oven set at 65 °C until they reached a constant weight. After this process, the samples were ground and sieved, and then measured. Sample (0.5 g) was added to a 200 mL beaker containing deionized water (80 mL), stirred using a magnetic stirrer for 5 min, and then nitric acid (1 + 5, 50 mL) was added, and the mixture was then stirred for a further 25 min. Immediately, the sample was placed under a vacuum and filtered through a glass sand core funnel (40 mL, G3), dried at 105 °C to a constant weight, and the remaining insoluble slag in the magneton and beaker was cleaned using anhydrous ethanol. The slag was then transferred to the sand core funnel, which was then placed in an oven set to 105 °C and repeatedly dried and weighed until a constant weight was achieved. The degree of reaction, *α_SL_*, was measured, and calculated using the following Equation (2):(2)$$\alpha_{\mathrm{SL}}=1\;-\;\frac{\frac{W}{1-\mathrm{Wn}}{-f}_{c}W_{c,e}}{f_{\mathrm{SL}}W_{s,e}}$$
where *α_SL_* is the degree of reaction of the GFS in %, *W* is the mass fraction of the blended cement at the specified age after dissolution in nitric acid in %, *W_n_* is the non-evaporated water content of the blended cement at the specified age in %, *W_c,e_* is the mass fraction of the cement hardened paste after dissolution in nitric acid in %, *W_s,e_* is the mass fraction of the GFS raw material after dissolution in nitric acid in %*, f_c_* is the mass fraction of cement in the blended cement in %, and *f_SL_* is the mass fraction of GFS in the blended cement in %.

(4) Microscopic morphology and composition of the hardened paste

Test samples at the specified age were crushed and terminated using anhydrous ethanol for 1 d. The prepared samples were then dried in a vacuum drying oven set at 65 °C for 24 h and analyzed by scanning electron microscopy-energy dispersive X-ray spectroscopy (SEM-EDS) and powder X-ray diffractometry (XRD). The samples were sprayed with gold before SEM measurements, and the microstructures of the pastes hardened for different hydration ages were observed by SEM (JSM-6700F). The compositions of the samples were analyzed by energy dispersive X-ray (EDX) spectroscopy. The samples were ground to a particle size of <80 μm and dried to a constant weight. The diffraction patterns of the crystal phases of the hydration products were measured by XRD, and the phase compositions of the target materials were analyzed using CuKα radiation, at a voltage of 40 kV, current of 30 mA, over a 2*θ* scan range of 5°–70° at a scanning rate of 8°/min with a step size of 0.02.

(5) Mechanical properties testing

The compressive strengths of the hardened pastes and the cement mortar strength were measured using a pressing machine (BC-300D) over a period of 3–90 d according to the “Cement mortar strength test” (GB/T17671, Chinese standard method).

## 3. Results and Discussion

### 3.1. Analysis of the Dissolution Characteristics of the GFS Powder

As the hydration activity of GFS powder is affected by the degree of dissolution of its constituent Si and Al elements, the amounts of Si and Al dissolved in the GFS powder were measured under different alkaline conditions and temperatures, with the results shown in Figure 7. In the aqueous solution of the blank group, the dissolution rate of the Si and Al elements was low. As the GFS powder reacted, the Ca dissolved and reacted with Si^4+^ and Al^3+^ to produce a gel and calcium silicate. The concentrations of Si and Al in the solution system decreased slightly after 720 min. In a simulation of the cement environment (0.2 mol/L NaOH), the dissolution rates of Si and Al increased significantly with an extension of the reaction time, showing a rapid growth trend. Compared with the non-alkaline environment, the dissolution rates of Si and Al in the simulated cement environment were found to be significantly improved, with the improvement increasing in line with time. These results indicate that the alkaline environment promotes the hydration activity of the GFS powder. With an increase in NaOH concentration, the dissolution rates of Si and Al increased continuously, following the same trend. When the concentration of NaOH in the solution reached 0.6 mol/L, there were obvious increases in the dissolution rates of Si^4+^ and Al^3+^. After a reaction time of 720 min, the dissolution rates of Si and Al were 46.6 mg/L and 30.1 mg/L, respectively, values that are around 13- and 9-fold those of Si and Al in the non-alkaline environment, respectively. This shows that the increase in OH^−^ concentration had a positive effect on the alkaline activation of the GFS powder. Thus, the active components Al_2_O_3_ and SiO_2_ in the GFS powder are the source of its pozzolanic activity. In an alkaline solution, the Si–O and Al–O bonds break, the degree of polymerization of the Si–O–Al network decreases, and silicoaluminate is gradually dissolved [43,44]. These processes occur via the reactions shown in Equations (3) and (4). With an increase in the concentration of dissociated OH^−^ in solution, the rate of the forward reaction is faster, and the ion dissolution rate is higher.
(3)$${\mathrm{Al}}_{2}O_{3}\left(s\right){+2\mathrm{OH}}^{-}\left(\mathrm{aq}\right)\to {2\mathrm{AlO}}_{2}^{-}\left(\mathrm{aq}\right){+H}_{2}O$$
(4)$${\mathrm{SiO}}_{2}\left(s\right){+2\mathrm{OH}}^{-}\left(\mathrm{aq}\right)\to {\mathrm{SiO}}_{3}^{2-}\left(\mathrm{aq}\right){+H}_{2}O$$

Figure 7b shows the dissolution of GFS powder in the simulated cement environment at different temperatures. It can be seen from the data that temperature had a significant influence on the dissolution of Si and Al. Especially at 60 °C, the dissolution rates of Si and Al after 12 h were 179.7 mg/L and 118.1 mg/L, respectively, about 6–7 times higher than those measured at room temperature. The increase in temperature accelerated molecular movement, making the Si–O and Al–O bonds of active SiO_2_ and Al_2_O_3_ more likely to break, which means that more active molecules were present in the system. Thus, when the reaction rate was accelerated, the dissolution rates of Si and Al increased, indicating that an increase in temperature led to an improvement in the activity of the GFS powder.

### 3.2. Effect of GFS Powder on the Hydration Heat Release of the Blended Cement

Hydration heat release curves and the characteristics of the pure cement and blended cement samples containing GFS powder that exhibit different specific surface areas are shown in Figure 8 and Table 3, respectively. The hydration heat release process of the blended cement containing GFS powder was similar to that of pure cement, with an induction period of 1.3–2.5 h. With an increase in GFS powder content, the hydration heat release rate decreased gradually in the acceleration stage, and the end time of the induction stage was delayed. This delay effect may be related to the high specific surface area and dilution effect of the GFS powder. The induction period of cement hydration is mainly controlled by the nucleation stage. After the Ca^2+^ in the solution reaches the saturation concentration, it gradually precipitates, and the hydration products in the solution increase. The addition of GFS powder reduced the dissolved amount of Ca^2+^ in the cement system, delayed the time when Ca^2+^ in the solution reached the saturation concentration, and thus prolonged the hydration induction period. In addition, during its hydration, the surface of the GFS powder tended to adsorb a large amount of Ca^2+^, which also hindered the nucleation of Ca(OH)_2_ (abbreviated as CH) and reduced the concentration of Ca^2+^ in the pore solution. At the same time, the initial hydration activity of GFS powder was low, and its inert filling effect had a dilution effect, which increased the water-cement ratio during the actual reaction of the cement, thereby delaying its initial hydration. With an increase in the specific surface area of the GFS powder, there was an increase in the contact area between the particles and water, which promoted nucleation and accelerated the hydration of the cement. The rate of SiO_2_ and Al_2_O_3_ dissolved by GFS with higher specific surface area in alkali solution was accelerated, and the heat released by reaction with CH produced by cement hydration in unit time was increased. Thus, the main heat release peak appeared earlier, and the peak value increased.

According to Figure 8b, the cumulative heat release diagram of the hydration of the cement represents the hydration of the blended cement at any given point over 72 h. During this period, the cumulative heat release of pure cement was 267.67 J/g. The cumulative heat release values of cement mixed with 10%, 30%, and 50% GFS powder content were 91%, 73%, and 56% that of pure cement, respectively, and the heat release was higher than the theoretical total heat release of pure cement (heat release of cement × mass fraction of the cement). This indicates that the GFS powder was still partially involved in the hydration reaction despite its initial hydration activity being low. Ion dissolution experiments also showed that the alkaline environment of the cement hydration stimulated the activity of the GFS powder, which accelerated the overall degree of hydration of the system to a certain extent. At the same GFS content, with the increase in the specific surface area of the GFS powder, the total hydration heat release and hydration heat release rate showed an increasing trend. This indicates that grinding not only increased the fine particle content and the specific surface area of the particles, but also improved the hydration activity of the GFS powder.

### 3.3. Kinetics of the Initial Hydration of the Blended Cement Containing GFS Powder

Chemical kinetics provides information on the rate of the chemical reaction and the factors that affect the rate of reaction. The hydration reaction of cement and the blended cement could thus be explained from a chemical kinetics perspective. Many studies have analyzed and described the hydration process of blended cement. In this study, the Krstulović-Dabić model (K-D model) was used to determine the corresponding hydration dynamic parameters according to the K-D model [45,46,47], to understand the hydration mechanism of the GFS powder–cement composites.

In this model, the hydration of the blended cement can be divided into three processes: crystal nucleation and growth (NG), phase boundary reaction (I), and diffusion reaction (D) stages. It should be noted that all three stages can occur. Assuming that the slowest reaction plays a decisive role in the hydration process of the blended cement, it is necessary to calculate the hydration rates of the three stages. The initial stage of hydration of the cement may be controlled by the NG process, which gradually changes to an I or D process as the hydration of the cement continues. The final hydration process is thus dependent on the slowest reaction. The equations and differential equations of the three processes are as follows:

The equation for the NG process:(5)$${\left[-\ln(1-\;\alpha )\right]}^{\frac{1}{n}}=K_{1}(t\;-\;t_{0})={K^{’}}_{1}(t\;-\;t_{0})$$

The equation for the I process:(6)$${1\;-\;(1\;-\;\alpha )}^{\frac{1}{3\;}}=K_{2}r^{-1}(t\;-t_{0})={K^{’}}_{2}(t\;-t_{0})$$

The equation for the D process:(7)$${\left[1\;-\;{\left(1\;-\alpha \right)}^{\frac{1}{3}}\right]}^{2}=K_{3}r^{-2}\left(t\;-\;t_{0}\right)={K^{’}}_{3}\left(t\;-\;t_{0}\right)$$

The differential equation for the NG process:(8)$$\frac{d\alpha }{\mathrm{dt}}{=F}_{1}(\alpha )={K^{’}}_{1}{n(1\;-\;\alpha )[-\;\ln(1\;-\alpha )]}^{\frac{n-1}{n}}$$

The differential equation for the I process:(9)$$\frac{d\alpha }{\mathrm{dt}}{=F}_{2}{(\alpha )=3K’}_{2}{\left(1\;-\alpha \right)}^{\frac{2}{3}}$$

The differential equation for the D process:(10)$$\frac{d\alpha }{\mathrm{dt}}{=F}_{3}(\alpha )=\frac{{K’}_{3}\cdot {3(1-\alpha )}^{\frac{2}{3}}}{{2-2(1-\alpha )}^{\frac{1}{3}}}$$
where α is the degree of hydration, n is the geometric crystal growth index, t is the hydration time, t_0_ is the end time of the induction period, K_i_ is the reaction rate constant, r is the reaction particle diameter, K_i_′ is the apparent reaction rate constant, and F_i_(α) is a kinetic model function. To convert the obtained hydration heat data into the degree of hydration (α) and rate of hydration (dα/dt) required by the kinetics model, the hydration kinetics equation proposed by Knudson should be used:(11)$$\frac{1}{Q(t)}=\frac{1}{Q_{\max}\frac{t_{50}}{Q_{\max}{(t-t}_{0})}}$$
where Q(t) is the heat release calculated from the acceleration period; Q_max_ is the total heat released at the end of the theoretical hydration of the cementitious material; t_50_ is the half-life period, that is, the time required for the heat release of the cementitious material to reach 50% of the total heat release; and t − t_0_ is the hydration time calculated from the acceleration period. Therefore:(12)$$\alpha (t)=\frac{Q(t)}{Q_{\max}}$$

The calculation process of the kinetics model function can be divided into several steps: (1) The substitution of the processed heat of hydration data into Equation (11) and the calculation of Q_max_ by fitting. (2) The substitution of Q_max_ into Equation (12) to determine the actual degree of hydration α(t) and the substitution of α into Equations (5)–(7) to determine the values of n, K1′, K2′, and K3′. (3) The substitution of the values of n, K1′, K2′, and K3′ of the three hydration stages of NG, I, and D into Equations (8)–(10) to determine the relationship between dα/dt and α. According to the relationships between F1(α), F2(α), F3(α), and α, the hydration kinetics of the cementitious composite materials were analyzed.

Figure 9 shows the fitting curves of the hydration reaction rates of the blended cement. The *R*^2^_Total_ of the fitting process of the three hydration stages in all of the systems was >0.99, indicating that the pure cement and blended cement containing GFS powder basically conform to the multiple reaction mechanisms proposed by Krstulović-Dabić. According to the fitting results, the hydration of the blended cement does not proceed via a single reaction process, but rather a complex multiple reaction of NG, I, and D processes. In the initial stage of hydration, many tiny crystal nuclei of hydration products adhere to the surface of the GFS powder, and precipitate on the surface through the mechanism of dissolution and crystallization, providing nucleation sites and promoting cement hydration. These crystal nuclei then grow into a calcium silicate hydrate (C–S–H) gel and CH crystals. Therefore, this hydration stage is controlled by NG and good data fitting results are achieved. As the hydration process progresses, there is a gradual increase in the hydration products, and the NG process gradually becomes an I process. In the later stage of hydration, the hydration product layer thickens, the paste gradually becomes denser, and the space in which the hydration product can continue to grow becomes smaller. At this point, the control factors of hydration begin to be governed by process D. After this time, the continuation of the hydration reaction is dependent on the migration of a large amount of Ca and Si. Due to a low diffusion rate and large diffusion distance, the rate of reaction at this stage is very slow. It can also be seen from Figure 9 that the specific surface area and amount of GFS powder added to the cement do not change the hydration reaction mechanism of the blended cement. The hydration process can thus be determined to proceed in the following manner: NG→I→D. Figure 10 shows the possible hydration process model of the system [48,49].

Table 4 shows the hydration kinetics parameters of the blended cement. The *n* values of the blended cement containing GFS powder were all between 1 and 2, and lower than that of pure cement. With an increase in the content and specific surface area of the GFS powder, there was a gradual increase in the *n* values of the blended cement, indicating that the GFS powder had an impact on the geometric growth process of the cement during their hydration. As the NG process of the cement and blended cement is an autocatalytic reaction controlled by nucleation, this process exhibited the fastest rate of reaction among the three processes. The calculated data also showed that the apparent reaction *K*_1_′ rate constants of the systems were around four times faster than those of *K*_2_′, and fifteen times faster than those of *K*_3_′. Thus, it can be seen that the rate of reaction of the NG process is much faster than the I and D processes. The *K*_1_′ values of blended cement containing GFS powder were found to be lower than those of pure cement, indicating that GFS powder can delay cement hydration, which may be related to the inertness of the initial hydration of the GFS powder. The NG process of the cement-based materials systems mainly involved the rapid hydration of C_3_S and C_3_A in the clinker. With an increase in the GFS powder content, there was a decrease in the proportion of clinker in the blended cement, as well as a decrease in the Ca^2+^ content of the solution. Moreover, a large amount of Ca^2+^ was absorbed on the surface of the GFS powder, which reduced the concentration of Ca^2+^ in the pore solution and delayed the nucleation and crystallization of CH and C–S–H, thereby delaying the reaction and reducing the value of *K*_1_′. With an increase in the specific surface area of the GFS powder, there was a continuous increase in the *K*_i_’ values, which indicated that a higher specific surface area of the GFS powder could improve the chemical kinetic process of the cement system. From Table 4, it can be seen that *α*_1_ is the abscissa of the point of the intersection of F_1_(*α*) and F_2_(*α*) in Figure 8a, that is, the point at which the transition from the NG to I process occurred. *α*_2_ is the abscissa of the point of the intersection between F_2_(*α*) and F_3_(*α*) in Figure 8a, that is, the point at which the transition from the I to D process occurred. The *α*_1_ and *α*_2_ values of the blended cement containing GFS powder were higher than those of pure cement. This indicates that the inertness of the initial hydration of GFS powder led to a smoother reaction of the blended cement and a longer duration of the hydration process at each stage. Transformation of the reaction control mechanism occurs only upon an increase in the degree of hydration.

In summary, in the initial hydration of the cement, the content and specific surface area of GFS powder changed the heat release rate and total heat release of the hydration of cement by varying degrees, delaying the hydration of the cement, and reducing the heat release of the system. In the complex process of hydration, the addition of GFS powder thus affects every reaction process, thereby affecting the entire blended cement system. The hydration process of blended cement containing GFS powder with a content of ≤50% is therefore NG→I→D.

### 3.4. Degree of Reaction of Blended Cement and GFS Powder

The chemically-bound water in the hardened pastes of blended cement usually exists in the hydration products in the form of chemical bonds or hydrogen bonds, such as CH, C–S–H gel, AFt, etc. Its content thus reflects the relative content of hydration products; the greater the degree of hydration of the system, the higher the chemically-bound water content. The chemically-bound water content of blended cement of different ages with different GFS powder content with specific surface areas is shown in Figure 11.

As shown in Figure 11, the content of the chemically-bound water increased gradually with an extension in the hydration time, indicating that the degree of hydration of the materials increased in line with their age. Preliminary experiments showed that the activity release of the GFS powder was initially slow, and that its physical and chemical action affected the rate of the hydration of the cement. At a later stage of hydration, when the CH content was higher, the activation effect of the pozzolanic activity of the GFS powder was more pronounced, which continued to promote the degree of hydration of the materials. The CH generated by cement hydration gradually excited the active substances such as SiO_2_ and Al_2_O_3_ in GFS vitreous phase to generate C(A)SH. Thus, there was still a high increase in the bound water content of the materials over a period of 28 d–90 d. With an increase in the GFS powder content, the proportion of the cement in the blended cement decreased, the amount of hydration products in each phase of the system decreased, and the chemically-bound water content decreased. In particular, when the GFS powder content was 50%, the chemically-bound water content of GFS-I, GFS-II, and GFS-III decreased by around 20% compared with that of pure cement. With an increase in the specific surface area of the GFS powder, the chemically-bound water increased slightly when the GFS content remained constant. In particular, when the GFS content of GFS-III was 10%, its contents of chemically bound water after 3 d, 28 d, and 90 d were 101%, 103%, and 102% of pure cement, respectively. This indicates that a low GFS powder content with a high specific surface area improved the degree of hydration of the pure cement system to a certain extent.

Through the calculation of the chemically-bound water content of the blended cement and the experimental data of the selective dissolution of nitric acid, the degrees of reaction of GFS powder at different content and specific surface areas could be determined, as shown in Figure 12. The degree of reaction of GFS powder and blended cement had a positive correlation. At the early stage of cement hydration, the reaction of GFS powder was slow. The degree of reaction of the GFS powder in the system was < 5% after 3 d. With the increase in curing age, the degree of reaction of GFS powder deepened and increased significantly. When the content of GFS powder was 10%, the degree of reaction was higher. Among them, the response degrees of GFS-III samples after 3 days, 28 days and 90 days were 4.1%, 13.7% and 23.4%, respectively. With the increase in the content of GFS powder and the decrease in the specific surface area, the degree of reaction of GFS powder decreased in varying degrees. This shows that the degree of reaction of the low content of GFS powder was more complete in the hydration stage of the cement, and the improvement of specific surface area was helpful to deepen the degree of reaction of GFS powder.

The above results indicated that the GFS powder slowly reacted and exhibited low activity during the initial hydration stage. A high GFS powder content thus played a filling role in the initial hydration of the blended cement rather than a sufficient reaction, and the specific surface area of GFS powder had a great influence on its activity. In the later stage of curing, the amount of hydration products gradually increased, and the CH content gradually increased. The appropriate alkalinity dissociated the glass structure and induced the pozzolanic reaction of GFS powder. This again proved that the continuous alkaline environment in the pore solution and the high specific surface area of the powder could stimulate the activity of the GFS powder, which promoted the generation of hydration products and an improvement in the degree of hydration of the cement system.

### 3.5. Effect of GFS Powder on the Microstructures and Compositions of the Cement Hydration Products

The GFS powder not only affected the heat release and degree of hydration of the cement, but also affected the microstructure of the hardened pastes, which was mainly reflected in the composition, morphology, and distribution of the hydration products. SEM-EDS images and XRD patterns of the pure cement and blended cement containing GFS powder that had been allowed to harden for 3 d are shown in Figure 13.

In addition to the characteristic SiO_2_ peak observed in the mineral-phase composition of the GFS powder, the mineral composition and morphology of the hydration products of the blended cement were similar to those of the pure cement system. This indicates that the addition of GFS powder to the cement did not change the nature of its hydration products. The hydration products of the pure cement and 0.9Cement+0.1GFS-III samples were more abundant, and flake CH, flocculent C–S–H gel, and a small amount of needle-like AFt were observed on the surfaces of the samples. CH crystals with relatively complete crystallinity can also be observed in the XRD patterns of these samples, indicating that the pure cement and blended cement system underwent a certain degree of hydration, but their overall structures were not dense enough. Upon a gradual increase in the GFS powder content, there was a relative decrease in the cement content of the blended cement, with a significant decrease in CH flakes in the hydration products. There was more fibrous C–S–H in the systems, with some particles being in a state of partial hydration, just forming the embryonic form of the hydration products. In addition, there was a significant decrease in the intensity of the characteristic peaks of CH, unhydrated C_3_S, and C_2_S in the XRD patterns of the samples after their hardening for 3 d. This indicated that the hydration was slow and that there was a low degree of hydration. The mechanical force effect might lead to the destruction of some Si-O and Al-O polyhedral structures in GFS powder and the transition to the oligomeric state with higher activity. With an increase in the specific surface area of the GFS powder, its pozzolanic activity increased. The consumption of CH in the system increased, and there was an increase in the degree of hydration, resulting in a further decrease in the intensity of the characteristic CH peak in the XRD data.

Figure 14 shows the SEM and XRD patterns of pure cement and blended cement containing GFS powder after their hardening for 90 days. It can be seen that after 90 days, the densities of the hardened pastes were greatly improved compared with those of the pastes hardened for 3 d, the gap is reduced, and the hydration products are increased and evenly distributed. The C–S–H gel, CH, and other hydration products were closely intertwined and inlaid. The characteristic peaks of C_3_S and C_2_S weakened in intensity or even disappeared from the XRD data, with CH exhibiting a higher peak intensity and more complete crystallization, indicating an increased degree of hydration. At this time, the degree of reaction of the active components in the GFS powder was higher, and the degree of reaction of the blended cement was more complete. When the GFS powder content of the blended cement was high, the surface of the hardened paste was found to be relatively loose, the hydration products were dispersed, and an interlaced interface could be clearly observed. This indicates that a high content of GFS powder had a negative effect on the structures of the hydration products of the cement. Moreover, with an increase in the GFS powder content and specific surface area, there was a decrease in the intensity of the characteristic CH peak. This is related to the relative content of hydration products in the systems and the consumption of CH via the pozzolanic reaction of the GFS powder. In addition, there was a gradual decrease in the intensity of the characteristic peak of SiO_2_. This indicates that the physical milling process used to prepare the GFS powder weakened its crystal structure, created lattice defects, and increased its reactivity. This is consistent with the test results of the rate of hydration and the degree of reaction.

### 3.6. Effect of GFS Powder on the Strength of the Blended Cement and Mortar

Figure 15 shows the compressive strength of the pastes of the blended cement after 3 d, 7 d, 14 d, 28 d, and 90 d. At a GFS powder content of 10%, the compressive strength of the blended cement was higher than that of pure cement, especially in terms of the development in the strength of the blended cement after 7 days. The compressive strength of the 0.9C+0.1GFS-III sample reached 73.9 MPa and 81.3 MPa after 28 d and 90 d, respectively, which were around 108% and 111% of the strength values of pure cement at the same age. This indicated that the filling effect of a low GFS powder content and pozzolanic reaction had positive effects on the mechanical properties of the blended cement. When too much cement was replaced by a large amount of GFS powder, the amount of hydration products in the system decreased, and the bonding strength between hydration products weakened. At the same time, due to a decrease in environmental alkalinity, the activity of the GFS powder was not fully stimulated, leading to a decrease in the strength of the system. Thus, the higher the GFS powder content, the more obvious the decrease in strength. In addition, by comparing the compressive strength of cement mixed with GFS powder with different specific surface areas, it can be seen that the improvement in the specific surface area of the GFS powder promoted an improvement in the strength of the blended cement. Among the blended cement, the compressive strength of the 0.5C+0.5GFS-III sample after 28 d was around 30% higher than that of the 0.5C+0.5GFS-I sample. As an increase in the grinding time of the GFS powder, on the one hand, the ordered crystal structure of the GFS powder was partially destroyed, there was an increase in the number of lattice defects, and there was an improvement in the activity of the GFS powder, which was consistent with the XRD analysis results. On the other hand, when the content of active particles of <45 μm in size increased, there was an improvement in the particle morphology and pore structure of the GFS powder. The GFS powder thus had a more reasonable particle size distribution, and therefore exhibited a better filling effect in the process of cement hydration, resulting in materials with a denser structure and improved strength.

Figure 16 shows the compressive and flexural strength properties of cement mortar composites. Similar to the trends observed in the compressive strength of the pastes, the initial strength of the mortar system was low, with both the compressive and flexural strength increasing in line with an increase in the age of the material. At a GFS powder content of 10%, the compressive strength of the mortar system was improved to a certain extent. Compared with the mortar system of pure cement, the strength of the mortar 0.9C+0.1GFS-III sample after 7 d, 28 d, and 56 d increased by 17%, 7%, and 4%, respectively. However, the GFS powder had no obvious improvement effects on the flexural strength of the mortar system. The compressive strength and flexural strength of mortar were affected by many factors, such as the mineral composition of the blended system, the particle grading of the cementitious material, and the bonding strength of the interface transition zone between the cementitious material and the aggregate. The addition of GFS powder would change the relative content of the original C_3_S, C_3_A, C_4_AF and other components in the cement, reduce the Ca/Si ratio and the composition of C-(A)-S-H in the hydration products, and then affect the mechanical performance. With an increase in the GFS powder content, the compressive and flexural strength of the mortar system at different ages decreased, but the increase in the specific surface area of the GFS powder made up for the loss in strength to a certain extent. The results of the strength of the paste and mortar showed that a GFS powder content of 10% resulted in optimal mechanical properties of the cement-based materials system being observed, particularly in the strength of the paste at a later stage.

## 4. Conclusions

In this study, the reactivity characteristics of GFS powder and the influence they have on the hydration process and hardening characteristics of cement were studied. The initial hydration kinetics characteristics, hydration reaction process, morphology, microstructure evolution of the hydration products, and the mechanical strength development of blended cement with different specific surface areas and different GFS powder content were compared and analyzed, with the following conclusions drawn:

(1) The main chemical composition and active components of the GFS powder are SiO_2_ and Al_2_O_3_, which endow it with a certain degree of pozzolanic activity. An alkaline environment was found to promote the dissolution of Si and Al in the GFS powder. When the concentration of OH^−^ in the solution reached 0.6 mol/L, the dissolution rates of Si and Al after 12 h were around 13- and 9-fold higher than in a non-alkaline environment. At 60 °C, the dissolution rates of Si and Al after 12 h were 179.7 mg/L and 118.1 mg/L, respectively, which are about 6–7-fold higher than those at room temperature. Amorphous phase active substances exhibited a better ionic dissolution rate and reaction driving force under the conditions of high alkalinity and temperature, which are the conditions for GFS powder to be used as a cement mineral admixture.

(2) The initial hydration activity of the GFS powder was low, which prolonged the hydration induction period of the cement system, thus reducing the hydration heat release rate and total heat release. The increase in the specific surface area reduced this trend to a certain extent and improved the chemical kinetic process of the cement system. The specific surface area and GFS powder content did not have any effect on the hydration mechanism of the cement, according to the Krstulović-Dabić kinetics model. The hydration process involved crystal nucleation and growth (NG), phase boundary reaction (I), and diffusion reaction (D) stages. Of these processes, the rate of crystallization nucleation was much faster than the rate of phase boundary reaction and diffusion.

(3) The activation effect of GFS powder was better in the later stage of cement hydration. The increase in specific surface area and the decrease in dosage could improve the degree of reaction of GFS powder. The degree of reaction of GFS powder and blended cement had a positive correlation. GFS powder, when present in the cement at a low content (10%) with a high specific surface area (463 m^2^/kg), improved the degree of hydration of the pure cement system to a certain extent, and the degree of reaction of GFS powder reached 23.4% after 90 d.

(4) GFS powder did not change the type of the cement hydration products, with the main hydration products being Ca(OH)_2_, AFt, and C–S–H gel. If GFS powder replaced too much cement, it would lead to fewer hydration products, a loose structure, and an obvious decrease in the mechanical strength of the cement paste and mortar system. The increase in the specific surface area of the GFS powder made up for the loss of strength of the blended cement to a certain extent. The compressive strength of the 0.9C+0.1GFS-III paste and mortar system at each age was higher than that of pure cement, but the effect of the GFS on the flexural strength of the mortar system was not obvious.

## Figures and Tables

**Figure 1 materials-16-01922-f001:**
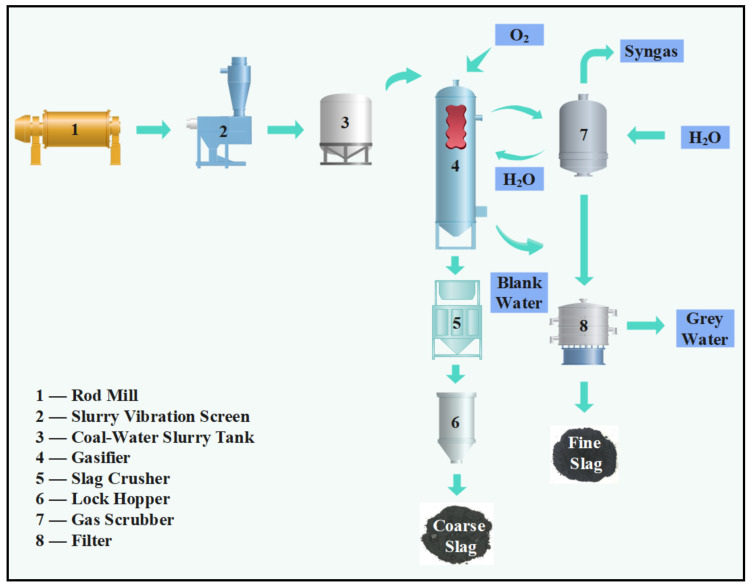
Schematic diagram of the coal gasification process [13].

**Figure 2 materials-16-01922-f002:**
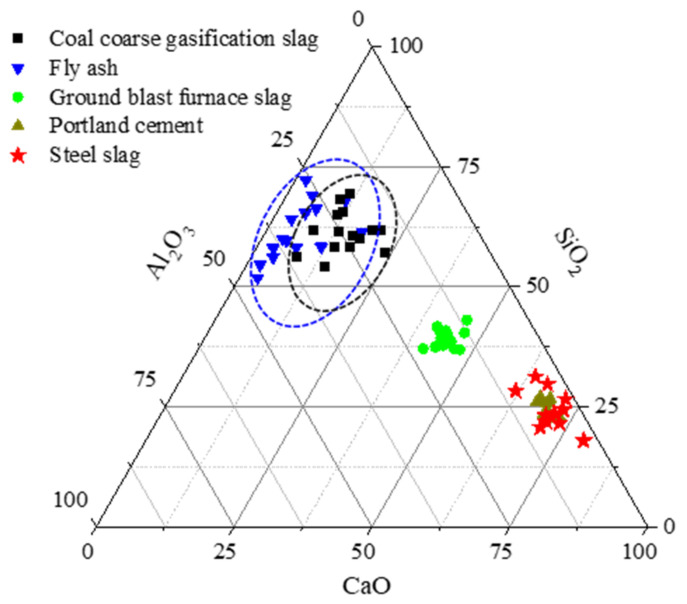
Composition comparison of different mineral admixtures.

**Figure 3 materials-16-01922-f003:**
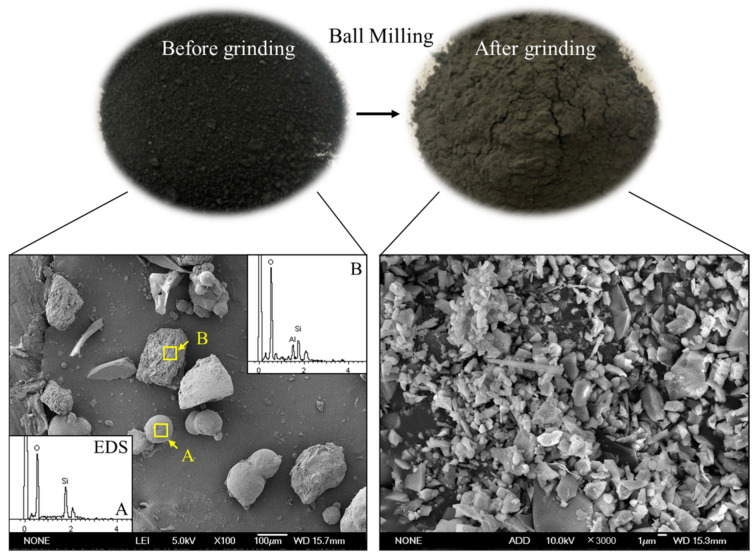
Particle morphology of the GFS powder.

**Figure 4 materials-16-01922-f004:**
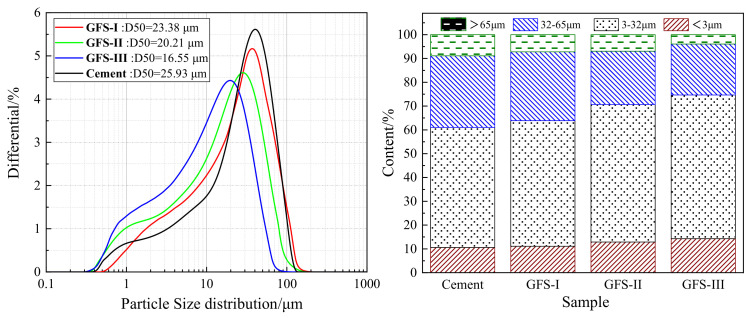
Particle size distribution of the GFS powder.

**Figure 5 materials-16-01922-f005:**
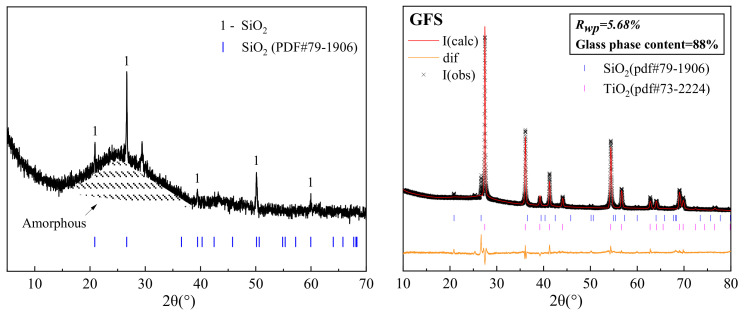
Analysis of the XRD (**left**) and QXRD (**right**) pattern of the GFS powder.

**Figure 6 materials-16-01922-f006:**
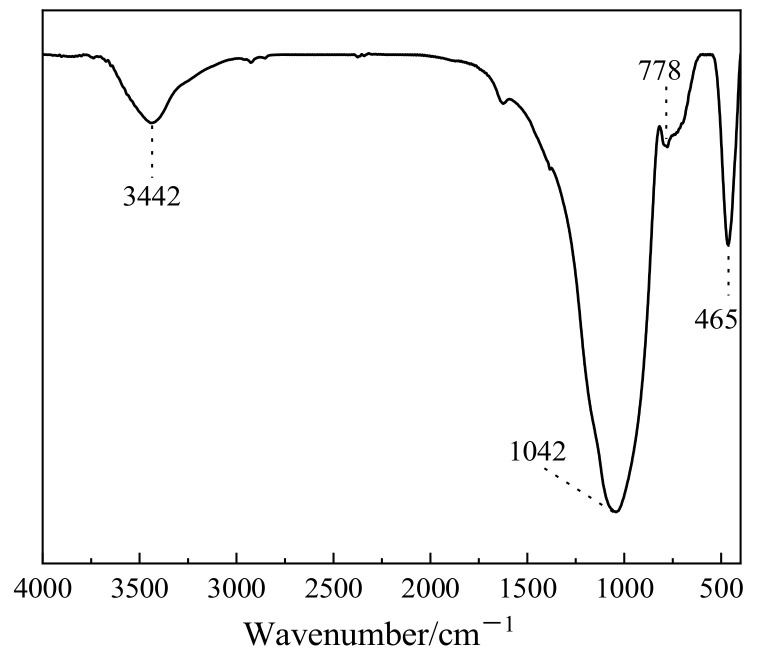
FTIR spectrum of the GFS powder.

**Figure 7 materials-16-01922-f007:**
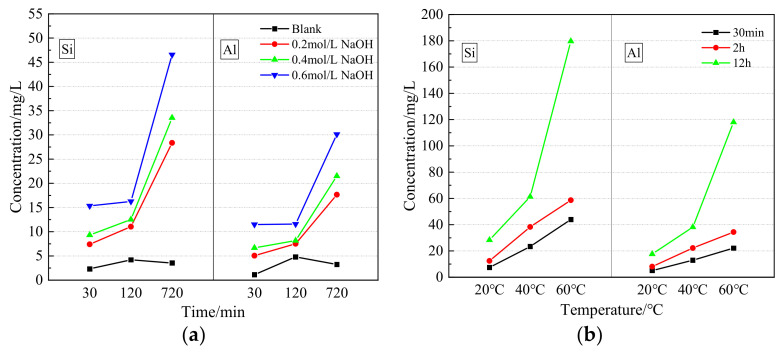
Si and Al dissolution characteristics of the GFS powder. (**a**) Dissolution of Si and Al in GFS powder at different temperatures, (**b**) Dissolution of Si and Al in GFS powder under different alkaline conditions.

**Figure 8 materials-16-01922-f008:**
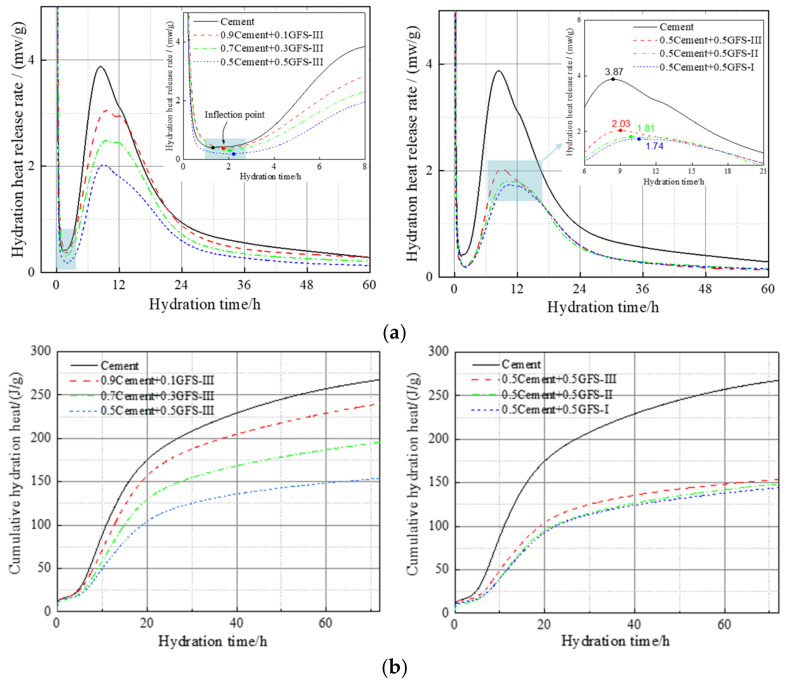
Hydration heat release rate (**a**) and cumulative hydration heat (**b**) of the different cement pastes.

**Figure 9 materials-16-01922-f009:**
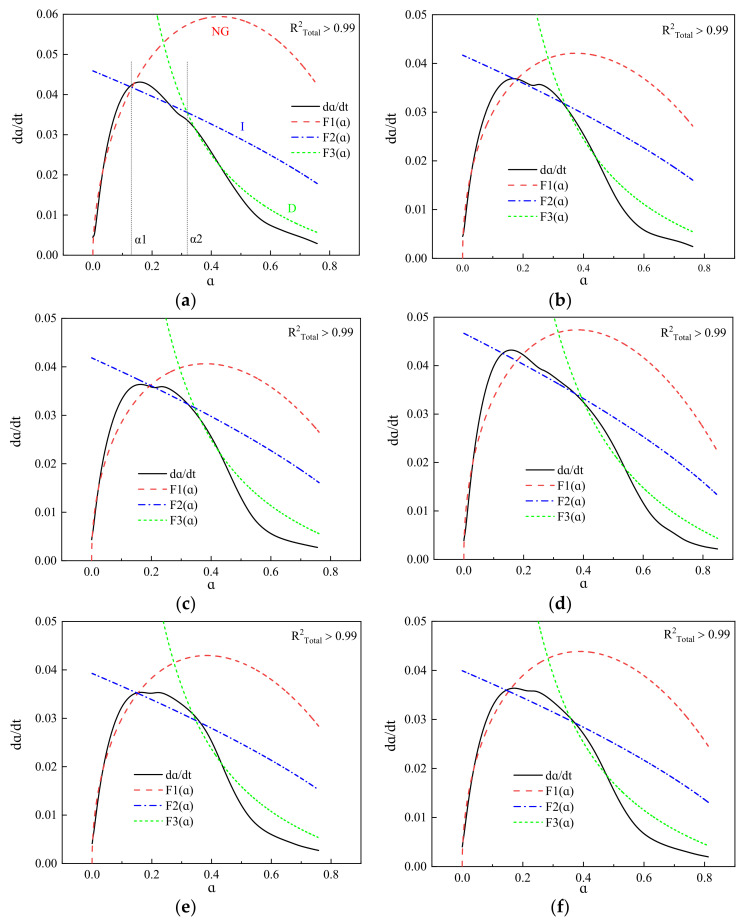
Fitting curve of composite cementing material reaction rate. (**a**) Cement, (**b**) 0.9Cement+0.1GFS-III, (**c**) 0.7Cement+0.3GFS-III, (**d**) 0.5Cement+0.5GFS-III, (**e**) 0.5Cement+0.5GFS-II, (**f**) 0.5Cement+0.5GFS-I.

**Figure 10 materials-16-01922-f010:**
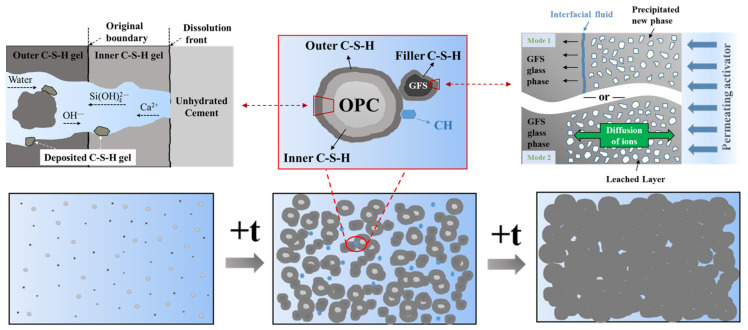
The hydration process model.

**Figure 11 materials-16-01922-f011:**
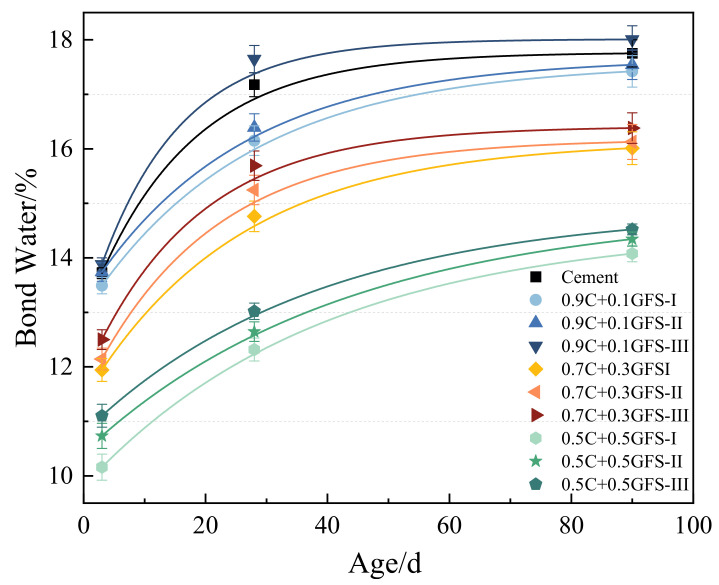
The chemically-bound water content of the blended cement.

**Figure 12 materials-16-01922-f012:**
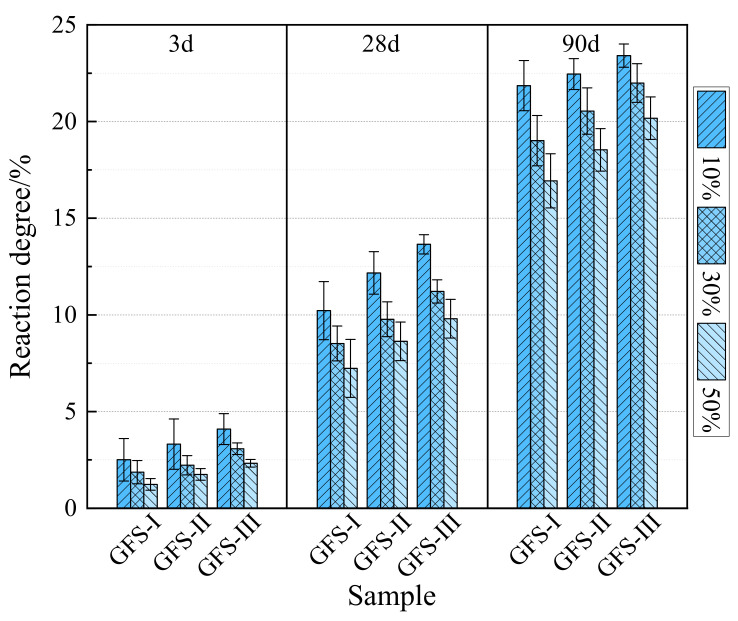
Degrees of reaction and growth rates of the GFS powder.

**Figure 13 materials-16-01922-f013:**
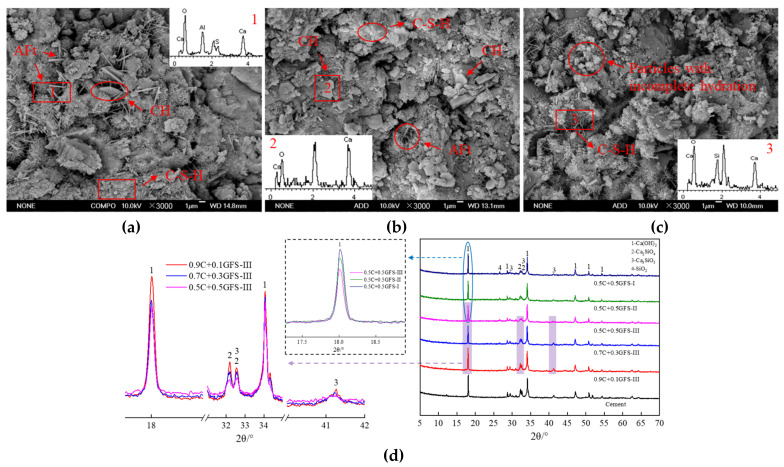
Microstructures and compositions of the hydrated products of the GFS-blended cement pastes after their hardening for 3 d. (**a**) Cement, (**b**) 0.9Cement+0.1GFS-III, (**c**) 0.5Cement+0.5GFS-III, (**d**) XRD patterns of the materials hardened for 3 d.

**Figure 14 materials-16-01922-f014:**
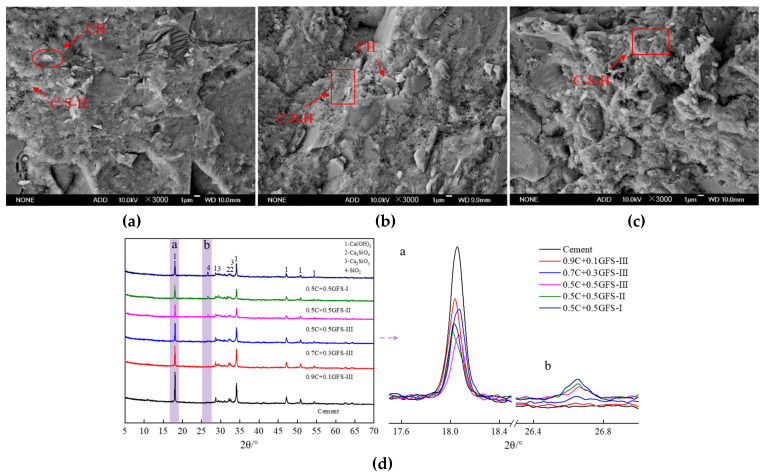
Microstructures and compositions of the hydrated products of GFS-blended cement pastes after their hardening for 90 d. (**a**) Cement, (**b**) 0.9Cement+0.1GFS-III, (**c**) 0.5Cement+0.5GFS-III, (**d**) XRD patterns of the blended cement hardened over 90 d.

**Figure 15 materials-16-01922-f015:**
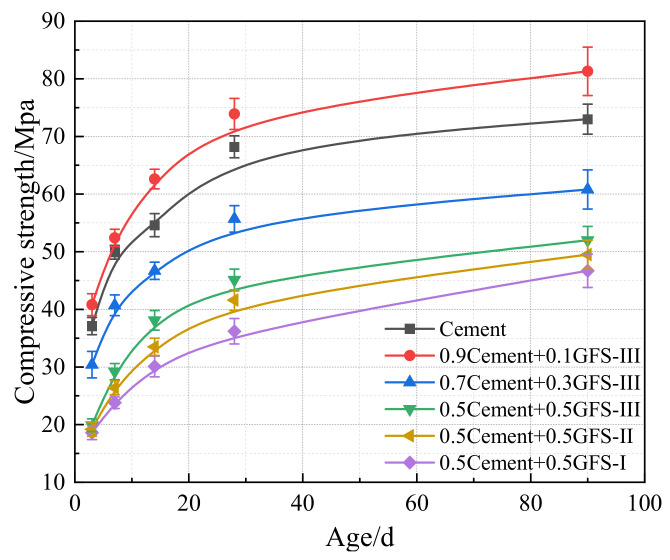
Compressive strength of hardened pastes of the blended cement at different ages.

**Figure 16 materials-16-01922-f016:**
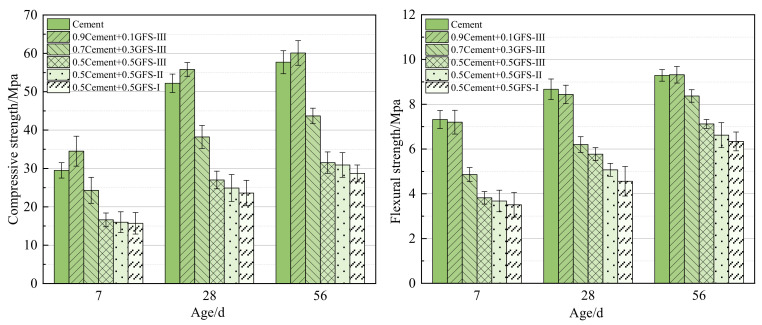
Compressive and flexural strength properties of the cement mortar composites.

**Table 1 materials-16-01922-t001:** Chemical compositions (/%) of cement and GFS.

	SiO_2_	Al_2_O_3_	Fe_2_O_3_	CaO	MgO	SO_3_	Loss on Ignition
Cement	20.78	5.01	3.50	63.93	2.01	2.21	1.80
GFS	54.31	18.76	6.37	8.89	2.00	3.12	2.95

**Table 2 materials-16-01922-t002:** The composition of cement and blended cement, expressed as a mix ratio in /%.

NO.	Sample	Cement	GFS-I	GFS-II	GFS-III	w/b
1	Cement	100				0.35
2	0.9Cement+0.1GFS-III	90			10	0.35
3	0.7Cement+0.3GFS-III	70			30	0.35
4	0.5Cement+0.5GFS-III	50			50	0.35
5	0.9Cement+0.1GFS-II	90		10		0.35
6	0.7Cement+0.3GFS-II	70		30		0.35
7	0.5Cement+0.5GFS-II	50		50		0.35
8	0.9Cement+0.1GFS-I	90	10			0.35
9	0.7Cement+0.3GFS-I	70	30			0.35
10	0.5Cement+0.5GFS-I	50	50			0.35

**Table 3 materials-16-01922-t003:** Hydration characteristics and cumulative heats of different cement pastes.

Samples	End Time of the Induction Period /h	Total Heat Release /J/g	Total Theoretical Heat Release /J/g
Cement	1.36	267.67	267.67
0.5Cement+0.5GFS-I	2.48	143.92	133.83
0.5Cement+0.5GFS-II	2.41	145.77	133.83
0.9Cement+0.1GFS-III	1.89	243.66	240.90
0.7Cement+0.3GFS-III	2.08	195.53	187.37
0.5Cement+0.5GFS-III	2.21	153.80	133.83

Note: total theoretical heat release = heat release of cement × mass fraction of the cement.

**Table 4 materials-16-01922-t004:** Kinetic parameters of gasification slag composite cementitious material.

Group	*n*	*K*_1_′	*K*_2_′	*K*_3_′	Hydration Mechanism	*α* _1_	*α* _2_
Cement	2.272	0.06335	0.0153	0.00367	NG-I-D	0.13	0.32
0.9Cement+0.1GFS-III	1.990	0.05681	0.0139	0.00359	NG-I-D	0.18	0.34
0.7Cement+0.3GFS-III	1.926	0.05323	0.01425	0.00366	NG-I-D	0.20	0.35
0.5Cement+0.5GFS-III	1.914	0.05209	0.01556	0.00478	NG-I-D	0.19	0.39
0.5Cement+0.5GFS-II	1.939	0.05177	0.01339	0.00397	NG-I-D	0.15	0.35
0.5Cement+0.5GFS-I	1.961	0.05084	0.01331	0.00371	NG-I-D	0.15	0.36

## Data Availability

Not applicable.

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
