# Peer review of "Utilization of Gasification Coarse Slag Powder as Cement Partial Replacement: Hydration Kinetics Characteristics, Microstructure and Hardening Properties"

_materials, 2023, doi:10.3390/ma16051922_

Round 1

Reviewer 1 Report

1) Many typo errors

Ground gasification coarse slag was represented in title inside the script it was mentioned as Coal gasification coarse slag (GFS). Authors needs to clarify the same.

2) Short forms are used with out proper representation.

3) Many statements are not not properly represented with out reason it was mentioned.

4) Research Gap ,Novelty and How this study will be helpful to the society needs to be highlighted.

5) Sample size and sampling procedure was not well explained

6) English and grammer needs to be improved

7) Many references old and recent references with respect to the context of the study needs to be added.

8) Discussion part must be strengthened

9) How you study helpful to the society?

10) What initiative you will take to commercialize your study outcome

Reviewer 2 Report

Executive Summary

Conversion of coal into oil or gas is an urgent task in the modern chemical industry. This produces large quantities of solid slag, which can potentially be used in the production of building materials. Research in this direction has been going on for several years. However, the full-scale use of slags from coal gasification requires comprehensive research. These studies should include both mechanical tests of cements containing coal gasification slag and physicochemical studies of its influence on hydration and hardening of cement paste. This allows us to talk about the relevance of the task.

Comments on the general concept

Statement of the problem.

The aim of the article is to systematically study the characteristics of the reactivity of GFS and its influence on the mechanism of cement hydration and hardening of the material. In particular, to deepen the study of the mechanism of influence of HPS powder on cement hydration kinetics and to quantify the degree of reaction of GFS powder in the process of cement hydration. This study discusses the properties of ground GFS powder, and the effect of GFS powder with different specific surface area and its content in mixed cement on material properties. The degree of hydration and cement hardening characteristics in terms of hydration kinetics, hydration reaction process, hydration product morphology and microstructure evolution as well as the development of mechanical strength of paste and mortar were investigated.

The section "Materials and methods" describes in sufficient detail the preparation of samples, and their research using modern physico-chemical methods of analysis. The results of the study are presented clearly, contribute to the scientific field of research, complete and correctly formulated. The conclusions are supported by the analysis of the results, provide answers to the research questions.

The disadvantages noted.

The authors used NaOH solution (0.2 mol/L) to determine the dissolution rate of Si and Al when modeling the cement medium. In this case, the dissolution rate of Si and Al significantly increased with increasing reaction time, demonstrating a tendency of rapid growth. For model systems, this can be used to some approximation. It is desirable to carry out studies to determine the dissolution rate of Si and Al in saturated solution of Ca(OH)2.

The noted disadvantages are not of principle nature. They do not affect the scientific content of the article and can be the subject of the following studies.

Conclusion

The scientific article is quite informative for specialists and is of interest for researchers dealing with the problem of supplementary cementitious materials (SCMs) and, in particular, the use of slag from coal gasification.

The article can be recommended for publication

Reviewer 3 Report

The comprehensive results and discussion is presented in the manuscript “Utilization of ground gasification coarse slag as cement partial replacement: hydration kinetics characteristics, microstructure and hardening properties”. The manuscript is well structured. There are several comments. 

1.  Abstract: It seems to me that these two conclusions contradict each other “The specific surface area of GFS powder and its content did not affect the reaction mechanism of cement” and “Higher specific surface area of the GFS powder could improve the chemical kinetic process of cement system.”

2. Of interest are the results on determining the workability of freshly prepared mixtures with an increase of the amount of finely ground slag since the water-to-binder ratio was constant and equal to 0.35.

3. What is the practical applicability of the results of this study?

Reviewer 4 Report

Dear authors,

I congratulate you for the research carried out in this work, for the structuring of the work and for the novelty.

I recommend you to correct the references according to the editorial instructions of the journal.

1.          Author 1, A.B.; Author 2, C.D. Title of the article. Abbreviated Journal Name Year, Volume, page range.

Reviewer 5 Report

The paper studies several properties and characteristics of Gasification Coarse Slag (GFS) as a partial replacement of cement.

The paper gives a good theoretical background to the topic, describes the materials and methods very clearly and the conclusions are based on the results. The paper is generally well written. The only thing I am missing is a general discussion where you compare GFS to other SCMs in terms of the properties studied in your paper.

Round 2

Reviewer 1 Report

I appreciate the author efforts in addressing the comments.

1) Many equations are adopted in the research study. It needs to be briefed and equation  number must be given.

2) Some sentences are too lengthy. It can be break down in to two.
